# Impact of a Femoral Fracture on Outcome after Traumatic Brain Injury—A Matched-Pair Analysis of the TraumaRegister DGU^®^

**DOI:** 10.3390/jcm12113802

**Published:** 2023-05-31

**Authors:** Mila M. Paul, Hannah J. Mieden, Rolf Lefering, Eva K. Kupczyk, Martin C. Jordan, Fabian Gilbert, Rainer H. Meffert, Anna-Leena Sirén, Stefanie Hoelscher-Doht

**Affiliations:** 1Department of Orthopedic Trauma, Hand, Plastic and Reconstructive Surgery, University Hospital of Würzburg, 97080 Würzburg, Germanyhoelscher_s@ukw.de (S.H.-D.); 2Department of Neurophysiology, Institute for Physiology, Julius-Maximilians-University Würzburg, 97070 Würzburg, Germany; 3Department of Neurosurgery, University Hospital of Würzburg, 97080 Würzburg, Germany; 4Institute for Research in Operative Medicine (IFOM), University of Witten/Herdecke, 51109 Cologne, Germany; rolf.lefering@uni-wh.de; 5LMU Klinikum Campus Innenstadt, University of München, 80336 Munich, Germany

**Keywords:** traumatic brain injury, femoral fracture, damage control orthopedics, mortality

## Abstract

Traumatic brain injury (TBI) is the leading cause of death and disability in polytrauma and is often accompanied by concomitant injuries. We conducted a retrospective matched-pair analysis of data from a 10-year period from the multicenter database TraumaRegister DGU^®^ to analyze the impact of a concomitant femoral fracture on the outcome of TBI patients. A total of 4508 patients with moderate to critical TBI were included and matched by severity of TBI, American Society of Anesthesiologists (ASA) risk classification, initial Glasgow Coma Scale (GCS), age, and sex. Patients who suffered combined TBI and femoral fracture showed increased mortality and worse outcome at the time of discharge, a higher chance of multi-organ failure, and a rate of neurosurgical intervention. Especially those with moderate TBI showed enhanced in-hospital mortality when presenting with a concomitant femoral fracture (*p* = 0.037). The choice of fracture treatment (damage control orthopedics vs. early total care) did not impact mortality. In summary, patients with combined TBI and femoral fracture have higher mortality, more in-hospital complications, an increased need for neurosurgical intervention, and inferior outcome compared to patients with TBI solely. More investigations are needed to decipher the pathophysiological consequences of a long-bone fracture on the outcome after TBI.

## 1. Introduction

Traumatic brain injury (TBI), usually caused by an external force to the head during road traffic, sports accidents, or a fall, is a leading cause of disability and mortality in high-income countries, especially among young individuals [1,2,3]. Approximately 37% of all injury-related deaths in the European Union can be attributed to TBI [4], and over 7.7 million TBI survivors live with a permanent disability, such as depression, progressive memory decline, inhibited motor function, and temporary or permanent cognitive dysfunction [5,6,7,8]. Even mild to moderate TBI can lead to long-term neurodegeneration, as demonstrated by animal studies and systematic analyses of sports-related concussions revealing neuronal death, dendritic degeneration, and synapse reduction not only in the zone of impact but also in more distant cortical regions [9,10,11]. Nearly 50% of all polytrauma patients admitted to German-speaking hospitals present with TBI, and about 1/3 of these patients suffer from severe, critical, or maximum concomitant extracranial injuries (abbreviated injury scale, AIS ≥ 3), such as thoracic trauma, abdominal trauma, or long bone fractures [12,13,14,15]. The impact of these injuries on patients’ outcomes obviously depends on their severity but also on the severity of the TBI. A meta-analysis of ~40,000 patients concluded that the influence of extracranial injuries strongly relied on TBI severity, showing a greater impact in patients with mild TBI [14]. Recently, implications of concomitant chest trauma and spleen lacerations on patients’ outcomes after TBI have been detected [16,17]. TBI and long bone fractures commonly occur together. Clinical evidence over the last decades suggests accelerated bone healing in TBI patients; however, the exact pathophysiology remains unclear [18,19]. Possible mechanisms involve pro-inflammatory cytokines and hormones [18,20,21,22,23] or the release of small extracellular vesicles in the hippocampus targeting osteoprogenitors [24]. In addition, patient-related and animal-model studies revealed that concomitant long-bone fractures worsen neurological and behavioral recovery after TBI [19,25,26]. In these patients, secondary brain damage is promoted by microvascular leakage, leukocyte–endothelial interactions in the penumbra [27,28], and the promotion of neuroinflammatory responses [22]. Despite the immense effort, the development of therapeutical strategies for multi-traumatized patients with TBI and concomitant long bone fractures remains challenging. 

The aim of this study was to analyze the impact of a concomitant femoral fracture on the outcome and mortality of TBI patients in a large cohort. Therefore, we used the TraumaRegister DGU^®^ (TR-DGU), which provides detailed information on patient comorbidities and pre- and in-hospital management, including complications, an outcome at the time of discharge, and mortality. The detailed assessment allowed a systematic comparison of these parameters in patients with isolated TBI compared to patients with TBI and a concomitant femoral fracture. We compared trauma mechanisms in both patient groups as the correlation between mechanism and long-term outcome in TBI patients is still precarious [29]. Furthermore, we asked whether the type of surgical intervention for the treatment of the femoral fracture had an influence on patients’ outcome and mortality and compared patients treated with early total care (ETC) to those treated with damage control orthopedics (DCO) [30,31,32,33,34]. We hypothesized that a concomitant femoral fracture negatively influences outcome and mortality in TBI patients and further postulated that mortality can be decreased by using DCO for initial treatment. 

## 2. Materials and Methods

TraumaRegister DGU^®^ (TR-DGU). The TR-DGU of the German Trauma Society (Deutsche Gesellschaft für Unfallchirurgie, DGU) was founded in 1993 with the aim of documenting pseudonymized and standardized data of severely injured, multi-traumatized patients. Data within this multi-center database are collected prospectively in four consecutive time phases from the site of the accident until discharge from the hospital: (A) pre-hospital phase; (B) emergency room and initial surgery; (C) intensive care unit; and (D) discharge. Furthermore, the documentation includes detailed information on demographics, injury patterns, comorbidities, pre- and in-hospital management, a course on intensive care unit, and relevant laboratory findings, including data on transfusion and outcome of each individual. The inclusion criterion is admission to the hospital via the emergency room with subsequent ICU/ICM care or reaching the hospital with vital signs and dying before admission to ICU. The infrastructure for documentation, data management, and data analysis is provided by the AUC—Academy for Trauma Surgery (AUC—Akademie der Unfallchirurgie GmbH), a company affiliated with the German Trauma Society. The scientific leadership is provided by the Committee on Emergency Medicine, Intensive Care, and Trauma Management (Sektion NIS) of the German Trauma Society. The participating hospitals submit their data pseudonymized into a central database via a web-based application. Scientific data analysis is approved according to a peer review procedure laid down in the publication guideline of the TR-DGU. The participating hospitals are primarily located in Germany (90%), but a rising number of hospitals from other countries contribute data as well (at the moment, Austria, Belgium, China, Finland, Luxembourg, Slovenia, Switzerland, The Netherlands, and the United Arab Emirates). Currently, over 28,000 cases from almost 700 hospitals are entered into the database per year. Participation in the TR-DGU is voluntary. For hospitals associated with TraumaNetzwerk DGU^®^, however, the entry of at least a basic data set is obligatory for reasons of quality assurance. This study is in line with the publication guidelines of the TraumaRegister DGU^®^ and registered as TR-DGU project ID 2021-001.

Study Cohort Patients treated in German, Swiss, or Austrian trauma centers in the 10-year period between 2010 and 2019 were included in our study cohort (Figure 1). We included adult patients aged 16 or older. Patients that were transferred from or transferred to another hospital early (within <48 h) were excluded, as well as patients with a minor (AIS_Head_ = 1) or untreatable TBI (AIS_Head_ = 6). To reduce the confounding impact of further accompanying injuries, patients with severe, critical, or maximum injuries of body regions other than the head or lower extremities (AIS ≥ 3) were excluded from the study, except for femoral fractures. Missing values for the initial prehospital Glasgow Coma Scale (GCS) also lead to exclusion from the study cohort. Patients were divided into two groups: group 1, patients with an isolated TBI (AIS_Head_ = 2–5); and group 2, patients with TBI (AIS_Head_ = 2–5) and a concomitant femoral fracture. According to this definition, “isolated TBI” means that there was no other “relevant” injury (AIS ≥ 3) outside the head. The cohort was matched 3:1 (group 1: group 2) based on age, sex, American Society of Anesthesiologists (ASA) risk classification (1–2 or 3–4), the severity of the TBI, and prehospital unconsciousness (GCS 3–8 vs. 9–15). An overview of the selection and matching process is shown in Figure 1.

Variables extracted from the database for the matching process included basic demographic data such as sex, age, ASA risk classification, initial, i.e., pre-hospital measured GCS, and severity of the TBI. The primary outcome parameters were in-hospital mortality (within 6 h, within 24 h, and overall; Figure 2 and Figure 3, Table 1) and outcome at discharge from the hospital (Table 1). The overall mortality describes all patients that deceased during the primary hospital phase. Patients that passed away after discharge from the hospital are not included, as the data basis from the TR-DGU ends at this time point. According to the current TR-DGU guidelines, the outcomes were defined as death (1), unresponsive (2), severe disability (3), moderate disability (4), and good recovery (5). This variable replaced the previously used Glasgow Outcome Scale (GOS) in 2015 and referred to patients with any type of injury and not only TBI. The outcome was defined as unfavorable for values of 1–3 and as favorable for values of 4 and 5. Secondary outcome parameters included the days spent in the intensive care unit (ICU), the duration of stay within the hospital, the destination after discharge from the hospital (to home, to a rehabilitation facility, etc.), as well as complications during the stay such as multi-organ failure (MOF), sepsis, and thromboembolic events. Basic characteristics for the mechanism of injury, time from accident to hospital admission, the severity of concomitant injuries in other body regions than lower extremities, the overall ISS (Injury Severity Score), and the level of care (level 1, 2, or 3) were extracted from the database (Table 2). We compared different fracture treatments in group 2 patients with concomitant TBI and femoral fracture (Figure 4). Treatment options were as follows: (i) non-operative treatment (non-op) when no surgery was performed; (ii) damage control orthopedics (DCO) when the patient was treated with an external fixator first, and later osteosynthesis was switched to nail or plate; and (iii) early total care (ETC) when the patient received definitive treatment immediately. Further medical procedures, such as preclinical intubation or intubation in the ICU, blood transfusion, and neurosurgical intervention (Table 1), were analyzed. 

Statistical Methods Statistical analysis was performed using SPSS statistical software (SPSS Version 24.0, IBM Inc., Armonk, New York, NY, USA). Data are presented as mean with standard deviation (SD) for continuous variables or as median with inter-quartile range (IQR) and as absolute numbers or percentages for categorical variables. The Chi-square-test or Mann–Whitney U-test were used to compare categorical and ordinal/continuous variables, respectively. Statistical significance was defined as a *p*-value < 0.05.

## 3. Results 

After applying inclusion and exclusion criteria, the successfully matched patient cohort consisted of 4,508 cases (Figure 1). Matching criteria included age, sex, ASA risk classification, and TBI severity defined by the AIS and GCS. The dominant sex in the study cohort was male (63%), and the mean age was ~55 years. Table 2 summarizes the basic characteristics of the matched study groups. As expected, the ISS was lower in group 1 patients (isolated TBI) than in group 2 patients (concomitant TBI and femoral fracture). Whereas the time from accident to hospital admission was comparable in both groups, the mechanism of injury varied substantially. The most common injury mechanism in patients with isolated TBI was a low fall, whereas it was a traffic accident in patients with combined TBI and femoral fracture (Table 2). Interestingly, all types of traffic accidents, except bicycle accidents, led to an increased number of combined traumata. Comparing the level of care (indicated by the current status of certified trauma centers, Weißbuch Schwerverletztenversorgung, https://www.traumanetzwerk-dgu.de/infos-downloads ((assessed on 1 September 2020).) both groups were mostly treated in level 1 trauma centers (supra-regional); however, group 1 patients with isolated TBI were more often transported to level 2 trauma centers (regional) than group 2 patients (36.4% vs. 25.4%). Group 2 patients also suffered significantly more concomitant injuries AIS = 2 in other body regions apart from head and lower extremities than group 1 patients (thorax: 20.4% vs. 15.8%, *p* < 0.001; abdomen: 6.3% vs. 3.7%, *p* < 0.001; spine: 20.6% vs. 17.9%, *p* = 0.044; upper extremities: 32.1% vs. 21.8%, *p* < 0.001; pelvis: 10.0% vs. 4.2%, *p* < 0.001). 

Our analysis detected several differences in clinical outcomes between both groups (Table 1). Group 2 patients with combined TBI and femoral fracture remained in the intensive care unit (ICU) and in the hospital longer than group 1 patients with isolated TBI (*p* < 0.001). However, the need to stay in the ICU was the same (92.7% in group 1 and 92.9% in group 2). Group 2 patients with combined trauma were more likely to be intubated in the field (37% vs. 28.7%, *p* < 0.001) as well as during their stay in the ICU (55.4% vs. 44.1%, *p* < 0.001). Interestingly, whereas both groups showed similar complication rates for sepsis and thromboembolic events, the risk for multi-organ failure was significantly increased in group 2 patients with combined TBI and femoral fracture (*p* < 0.001, Table 1). Most remarkably, the need for any neurosurgical intervention during the in-hospital stay (i.e., trepanation, craniectomy, external ventricle drainage, implantation of a pressure sensor) was significantly increased in patients with combined trauma (22.4% in group 2 vs. 10.3% in group 1, *p* < 0.001, Table 1) suggesting a negative influence of the femoral fracture on TBI. Next, we asked whether the risk for an unfavorable outcome and mortality were increased after concomitant femoral fracture in TBI patients. Indeed, group 1 patients with isolated TBI had a higher chance of favorable outcomes at the time of discharge from the hospital and were more often discharged home than to a rehabilitation facility (Table 1). The overall mortality (i.e., comprising all patients that deceased during the hospital phase), as well as the 6-h and 24-h mortality, were increased in group 2 patients with combined trauma (Table 1). Interestingly, whereas the 6-h mortality was increased by more than 100% in patients with combined trauma, the 24-h mortality was only increased by ~50% (Table 1). Next, we compared mortality in both groups depending on the severity of the TBI. Whereas patients with combined trauma showed a significantly increased 6-h mortality when presenting with AIS_Head_ = 5 (14.3% vs. 6.9%, *p* = 0.005), there was no difference in the 24-h mortality in all AIS_Head_ = 2–5 subgroups. Remarkably, we found that patients presenting with moderate TBI (AIS_Head_ = 2) had a higher overall in-hospital mortality when suffering from a concomitant femoral fracture (*p* = 0.037, Figure 2). Next, we asked whether the surgical procedure selected for the treatment of the femoral fracture had an influence on in-hospital mortality. We distinguished between non-operative treatment, early total care, and damage control orthopedics (non-op, ETC, DCO, Figure 4). From all group 2 patients included in the study, 630 data sets were available for analysis. A total of 60.6% of the patients received ETC using nail or plate osteosynthesis, whereas only 19.2% of the patients were treated with DCO (Figure 4). Interestingly, the in-hospital mortality rate was similar in patients treated with ETC and DCO. Furthermore, we compared the observed outcome of group 1 with isolated TBI and group 2 with combined trauma, as well as their RISC II prognosis and the standardized mortality ratio describing the relation of expected to actual mortality (SMR, Figure 3). Even if the SMR was comparable in both groups (*p* = 0.569), the observed outcome was less favorable in group 2 patients with combined trauma and the RISC II prognosis was worse.

## 4. Discussion

We conducted a retrospective matched-pair analysis of 4508 patients collected in the TR-DGU over a ten-year period. We compared patients with isolated TBI to patients with TBI and concomitant femoral fractures to investigate the impact of the fractures on outcomes and mortality in TBI patients. Our analysis demonstrated that an additional femoral fracture increased the mortality of TBI patients, especially in cases of moderate TBI (AIS_Head_ = 2) and in the early posttraumatic phase in cases of critical TBI (AIS_Head_ = 5; Table 1; Figure 2 and Figure 3). In addition, our data show that a concomitant femoral fracture deteriorates the patient’s outcome at the time of discharge from the hospital and enhances the risk of MOF as well as the need for intubation or any neurosurgical intervention during the in-hospital period (Table 1). However, our data did not provide evidence for the superiority of DCO compared to ETC for femoral fracture fixation in TBI patients (Figure 4). The pathophysiology of TBI can typically be divided into two phases [35,36]: first, the direct biomechanical force to the skull and brain that may (also without skull fracture) lead to cell necrosis, axonal shearing, and disruption of the blood–brain barrier causing brain edema and bleeding. Whereas this “primary injury” is determined by the initial external impact and obviously cannot be influenced in retrospect, the so-called “secondary injury” subsequently evolves over minutes to months after exposure and is the result of cellular, metabolic, and inflammatory processes [36,37]. Among other aspects, neuroinflammation is of particular importance for secondary TBI pathophysiology and might be aggravated by systemic inflammatory responses in the case of concomitant extracranial injuries in polytraumatized patients. However, most clinical trials chasing for effective TBI treatments to prevent the devastating long-term consequences have investigated pure, monotrauma TBI, therefore disregarding the negative influence concomitant extracranial injuries might have. 

In recent years, the impact of additional long bone fractures in TBI patients has gained importance. Long bone fractures increase peripheral serum levels of pro-inflammatory cytokines and hormones (e.g., TNF-α, interleukin-1, growth factor), which pertain to the bone healing response but are also capable of deteriorating TBI pathophysiology and worsening patient’s outcome [22,23]. Furthermore, they might increase hemorrhagic shock, levels of reactive oxygen species, and the risk for embolic complications. Nevertheless, despite the high prevalence of multi-trauma patients suffering from TBI and additional extracranial injuries (about 50% of the TR-DGU patients present with TBI and 20–30% with fractures of the spine, pelvis or extremities [15]), the potential interactions and their relevance for TBI pathobiology remain insufficiently understood. However, several studies over the past years aimed at investigating the effect of concomitant extracranial injuries on patients’ outcomes after TBI [27]. Whereas it was previously postulated that additional extracranial injuries have little to no influence on TBI outcome [38,39], more recent work strongly suggested an increase in patient mortality and deterioration of functional outcomes [40,41]. However, most of these studies come along with several confounding factors, such as wide variations in injury locus and severity or age bias with younger polytrauma patients than for isolated TBI [27]. Using the TR-DGU database, we analyzed isolated TBI patients in comparison to patients with concomitant femoral fractures in a large patient cohort of over 4000 cases. As it has been shown that functional outcome in TBI patients linearly deteriorates with increasing age [42], it was of particular importance to use patient age as a matching criterion (compare Figure 1). In addition, to rule out confounding cycle-dependent hormonal changes, we also distinguished between the female and male sex. We also considered the ASA risk classification an important matching criterion because severe pre-existing conditions might heavily influence the risk for complications and mortality after single or multi-trauma. It is important to mention that we excluded all patients with severe, critical, or maximum extracranial injuries (AIS ≥ 3) apart from the femoral fracture from our study cohort, “losing” nearly 50% of the remaining dataset for the matching process (Figure 1). Nevertheless, this was essential to receive the best-adjusted dataset for solely analyzing the effect of the femoral fracture on patients’ outcomes and mortality after TBI without potentially confounding co-morbidities. The therefore corrected and still relatively large cohort of 4508 patients divided into 3:1 groups by the matching process still allows us to postulate a clearly negative impact of an additional femoral fracture on TBI patient mortality and outcome as well as complications, such as MOF or the need for intubation. Especially in cases of critical TBI (AIS_Head_ = 5), mortality was increased by TBI in the early posttraumatic phase (<6 h and <24 h, Table 1). One plausible explanation is the secondary injury following TBI, including cellular and inflammatory devastation; however, one cannot rule out that other factors, such as hemorrhagic shock, might raise mortality in these patients. Interestingly, the 6-h mortality was increased by more than 100% in patients with combined trauma (from 1.5% to 3.2%), whereas the 24-h mortality was only increased by ~50% (from 5.6% to 7.9%, Table 1). This indicates a relative reduction in mortality during the early time course after trauma. Interestingly, we found that the need for any neurosurgical procedure was significantly increased in patients with matched TBI severity when presenting with an additional femoral fracture (Table 1). This might indicate that the femoral fracture deteriorates TBI pathobiology increasing the secondary insult. Another strength of our work is the division into subgroups according to the severity of the TBI (AIS_Head_ = 2–5) albeit the obviously decreased case numbers in these subgroups (Figure 2). We found that, especially in cases of moderate TBI (AIS_Head_ = 2), the additional femoral fracture had a significant impact on patient mortality, which is in accordance with the previous findings (6.6% during the in-hospital stay in patients with combined trauma vs. 3.8% in isolated TBI patients; see also [14]). As patients with further injuries apart from TBI (AIS_Head_ = 2) and femoral fractures were excluded from the analysis, one of the two injuries (which are usually not lethal when occurring uniquely) or, more likely, the combination of both can be considered as fatal. It would be compelling to analyze this relation, especially regarding long-term functional consequences, in greater detail [41,43]. Unfortunately, the TR-DGU up to now does not provide additional information regarding functional patients’ outcomes beyond the time of discharge. In the future, further analyses using the newly emerging TBI module (Schädel–Hirn Trauma Modul) of the TR-DGU, including a functional follow-up 6 and 12 months after trauma, are appealing but beyond the scope of this manuscript. 

## 5. Limitations

Our study also presents some limitations [27]. First, the TraumaRegister DGU^®^ data assessment is always a retrospective analysis; however, it delivers detailed information about trauma mechanism, level of care, preclinical and inpatient treatment (times), adverse events, and outcome parameters [43]. Furthermore, it provides detailed information about the exact time of death (<6 h or <24 h after injury vs. in-hospital mortality) and enables the comparison between observed and predicted mortality as measured by the RISC II score (Figure 3). Second, considering age an important outcome factor after TBI [42], the small but significant age difference between both groups (53.9 vs. 56.1 years, *p* = 0.012, Table 1) needs to be mentioned. Third, the quality of the primary outcome parameter ‘outcome’ is debatable as it only inaccurately represents the neurological outcome of the patient and may itself be influenced by concomitant injuries [44]. As this variable changed in the TR-DGU in 2015 from GOS to “outcome” (more general and only derived from the GOS), the pooling of both datasets was necessary but might blur the results. Here, future outcome analyses in the above-mentioned TBI module of the TR-DGU might be promising advancements. In addition, we found no difference in the rate of thromboembolic events between both groups, although we consider especially fat embolism an important entity in TBI patients [45]. As the TR-DGU does not retrieve detailed information about this type of thromboembolic complication, analysis was not applicable, even though relevant to investigate in future work. Fourth, we hypothesized a positive influence of DCO in TBI patients with concomitant femoral fracture compared to ETC. Our data analysis could not confirm this hypothesis, as the in-hospital mortality rate was similar for patients treated with DCO and with ETC (Figure 4). However, the considerably smaller number of patients included in this analysis (630 cases) did not allow reasonable subgroup analysis regarding TBI severity. Interestingly, we also found no explicit correlation between the probability of DCO and the severity of TBI. It is still debated if the usage of DCO in TBI patients decreases the risk of potentiating secondary brain injury due to hemodynamic complications and inflammatory response [46,47]. Further work is needed to provide clear evidence that the surgical method of choice for femoral fracture fixation influences TBI pathobiology and patients’ outcomes.

## 6. Conclusions 

Our retrospective matched-pair analysis of a patient cohort with 4,508 cases from the TR-DGU over a ten-year period showed that patients with combined TBI and femoral fracture have higher mortality, an increased risk for in-hospital complications and neurosurgical intervention, and an inferior functional outcome compared to patients suffering from isolated TBI. In the future, detailed analysis of the central and peripheral pro-inflammatory processes promoted by an additional femoral fracture in TBI patients appears promising to decipher the complex pathophysiology in multi-trauma patients and possible therapeutical implications.

## Figures and Tables

**Figure 1 jcm-12-03802-f001:**
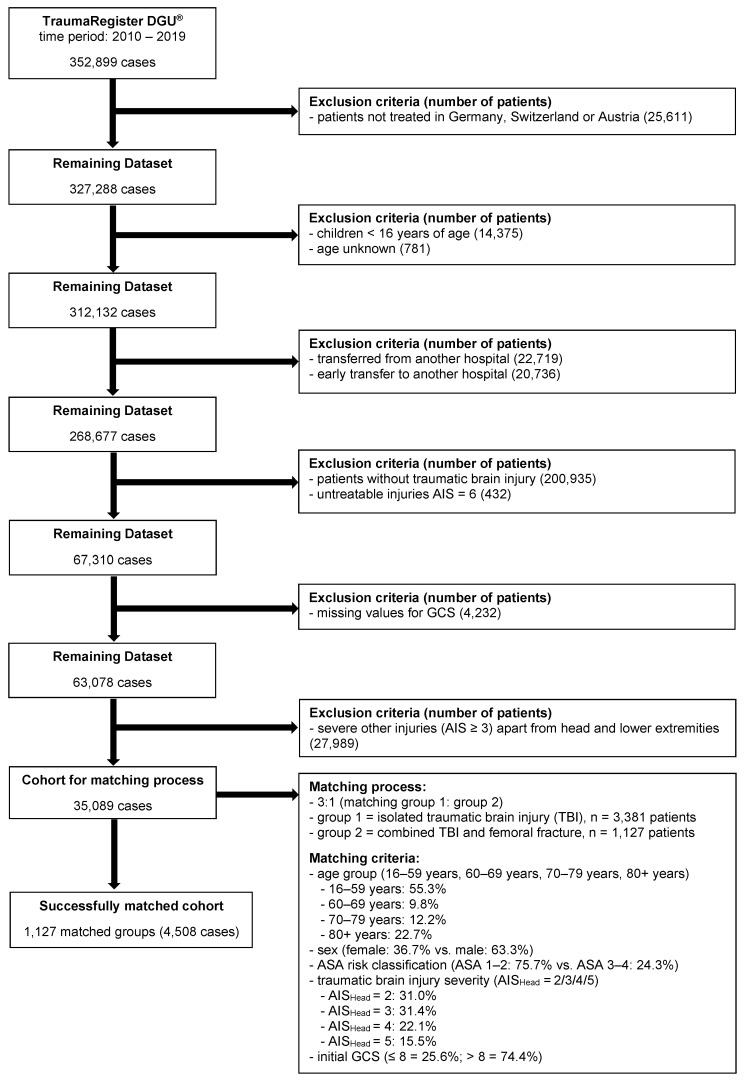
Flow chart describing case selection and matching process of the study cohort. AIS, abbreviated injury scale. ASA, American Society of Anesthesiologists. DGU, Deutsche Gesellschaft für Unfallchirurgie (German Trauma Society). GCS, Glasgow Coma Scale. TBI, traumatic brain injury.

**Figure 2 jcm-12-03802-f002:**
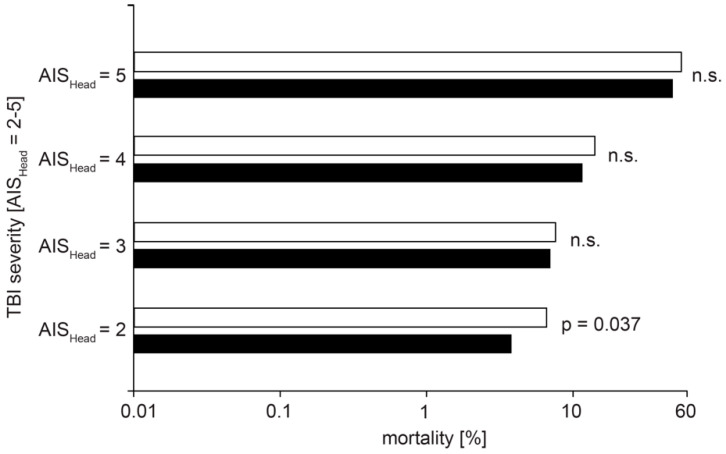
Concomitant femoral fracture increases mortality, especially in patients with moderate TBI. Diagram depicts relation between TBI severity (AIS_Head_ = 2, 3, 4, or 5; *y*-axis) and the overall mortality (*x*-axis; note logarithmic scale). Black bars indicate patients with isolated TBI (group 1), and white bars indicate patients with TBI and concomitant femoral fracture (group 2). Remarkably, in the AIS_Head_ = 2 group, mortality was significantly increased in patients with combined trauma. AIS, abbreviated injury scale. n.s., not significant. TBI, traumatic brain injury.

**Figure 3 jcm-12-03802-f003:**
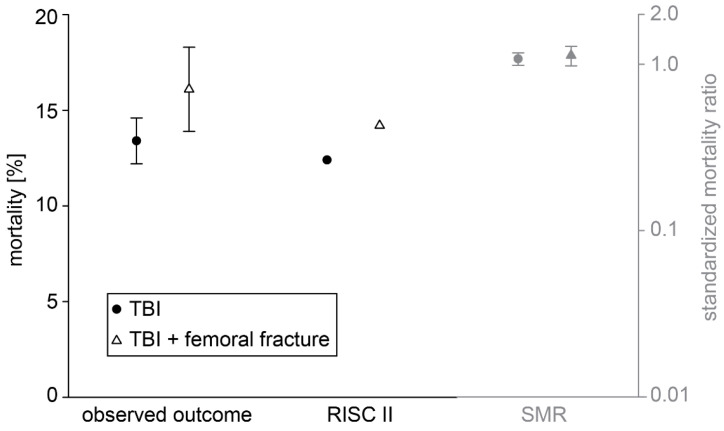
Increased observed and predicted mortality in patients with concomitant TBI and femoral fracture. Mean values (circles and triangles) and 95% confidence intervals (error bars) of observed and predicted mortality as calculated by the RISC II score (left scale, black) as well as the standardized mortality ratio (SMR, right scale, grey) in both groups. RISC II, revised injury severity classification score. SMR, standardized mortality ratio. TBI, traumatic brain injury.

**Figure 4 jcm-12-03802-f004:**
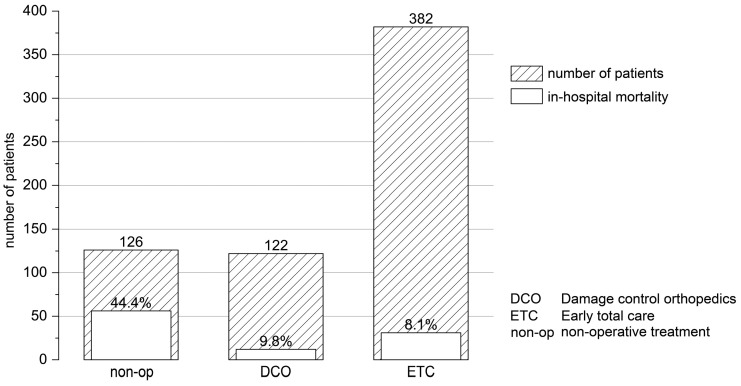
Surgical procedure and in-hospital mortality in patients with concomitant TBI and femoral fracture. For 630 of 1,127 group 2 patients presenting with both TBI and a concomitant femoral fracture, details regarding the applied surgical procedure for fracture care were available. In-hospital mortality was highest in patients receiving non-operative treatment (non-op) of their femoral fracture and comparable in patients receiving damage control (DCO) and early total care (ETC).

**Table 1 jcm-12-03802-t001:** Clinical Outcomes of the matched study groups (*n* = 4,508 patients). ICU, intensive care unit SD, standard deviation. TBI, traumatic brain injury.

Outcome Variable	Group 1 _TBI only_	Group 2 _TBI + femoral fracture_	*p*—Value
Number of Patients	3,381	1,127	
**Result presented as median and (IQR)**			
ICU length of stay in days	3 (1–7)	4 (2–11)	**<0.001**
Length of hospital stay in days *^1^	10 (6–18)	18 (10–27)	< 0.001
**Results presented as number of patients and in % per group**			
Neurosurgical intervention *^2^	268 (10.3)	186 (22.4)	**<0.001**
Multi-organ failure *^3^	214 (22.1)	177 (30.6)	**<0.001**
Sepsis *^4^	51 (5.4)	34 (5.9)	0.652
Thromboembolic event *^5^	21 (2.2)	15 (2.6)	0.568
Outcome			<0.001
Death	452 (13.4)	181 (16.1)	**0.024**
—within 6 h	52 (1.5)	36 (3.2)	**0.001**
—within 24 h	189 (5.6)	89 (7.9)	**0.005**
Unresponsive *^6^	62 (1.9)	25 (2.3)	-
Severe disability *^6^	281 (8.5)	115 (10.4)	-
Moderate disability *^6^	716 (21.7)	326 (29.4)	-
Good recovery *^6^	1792 (54.3)	461 (41.6)	-
Discharge from hospital			<0.001
—Discharge to home	1,667 (49.3)	419 (37.2)	-
—Discharge to rehabilitation facility	864 (25.6)	363 (32.2)	-
—Transfer to other hospital	281 (8.3)	99 (8.8)	-
—Other	117 (3.5)	65 (5.8)	-

*^1^ data only available for 3,349 group 1 and 1,124 group 2 patients. *^2^ data only available for 2,593 group 1 and 830 group 2 patients. *^3^ data only available for 968 of group 1 and 578 of group 2 patients. *^4^ data only available for 947 of group 1 and 573 of group 2 patients. *^5^ data only available for 967 of group 1 and 571 of group 2 patients. *^6^ data only available for 3,303 of group 1 and 1,108 of group 2 patients.

**Table 2 jcm-12-03802-t002:** Basic characteristics of the matched study groups (*n* = 4,508 patients). ISS, injury severity score. SD, standard deviation. TBI, traumatic brain injury.

Patient Characteristics	Group 1 _TBI only_	Group 2 _TBI + femoral fracture_	*p*—Value
Number of patients	3,381	1,127	
**Results presented as means (SD)**			
Age (years)	56.1 (22.7)	53.9 (25.4)	0.012
ISS	15.9 (7.5)	23.4 (7.6)	**<0.001**
Time from accident to hospital in minutes *^1^	61.0 (34.4)	68.4 (34.0)	**<0.001**
**Results presented as number of patients** **in % per group**			
Mechanism of injury *^2^			**<0.001**
—Traffic, overall	1,191 (36.4)	667 (59.8)	<0.001
—Traffic, car passenger	400 (12.2)	266 (23.9)	-
—Traffic, motorcyclists/socius	182 (5.6)	181 (16.2)	-
—Traffic, bicycle	430 (13.1)	103 (9.2)	-
—Traffic, pedestrian	159 (4.9)	101 (9.1)	-
—High fall ≥3 m	448 (13.7)	139 (12.5)	-
—Low fall <3 m	1,301 (39.7)	277 (24.8)	-
—Other	353 (10.8)	48 (4.3)	-
Level of care *^3^			<0.001
—Level 1 (supra-regional)	1,890 (55.9)	756 (67.1)	-
—Level 2 (regional)	1,230 (36.4)	286 (25.4)	-
—Level 3 (local)	261 (7.7)	85 (7.5)	-

*^1^ data only available for 2,832 group 1 and 991 of group 2 patients. *^2^ data only available for 3,273 of group 1 and 1,115 of group 2 patients. *^3^ level of care indicated by the current status of certified trauma centers according to the “Weißbuch Schwerverletztenversorgung”, https://www.traumanetzwerk-dgu.de/infos-downloads (assessed on 1 September 2020).

## Data Availability

The publication guideline of the TraumaRegister DGU^®^, at present, denies external access to raw data captured in the registry.

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
