# Peer review of "Impact of a Femoral Fracture on Outcome after Traumatic Brain Injury—A Matched-Pair Analysis of the TraumaRegister DGU®"

_jcm, 2023, doi:10.3390/jcm12113802_

Round 1
Reviewer 1 Report
This manuscript conducted a retrospective matched-pair analysis and found that patients with combined TBI and femoral fracture have a higher mortality. The results are interesting, and would provide insightful information to the readers of “Journal of Clinical Medicine”.
Major comment #1: Referring to “The overall mortality as well as the 6-hour and 24-hour mortality were increased in group 2 patients with combined trauma.” The overall mortality should be described in more detail.
Major comment #2: Referring to “The overall mortality as well as the 6-hour and 24-hour mortality were increased in group 2 patients with combined trauma.” The mortality after 24-hour seems to be decreased in group 2 patients with combined trauma, which was opposite to the overall mortality, 6-hour and 24-hour mortality. Did long term correlation and short term correlation were opposite? This is important and interesting.
Author Response
This manuscript conducted a retrospective matched-pair analysis and found that patients with combined TBI and femoral fracture have a higher mortality. The results are interesting, and would provide insightful information to the readers of “Journal of Clinical Medicine”.
We thank the reviewer for the helpful assessment of our manuscript and the positive evaluation. Addressing the interesting points raised by the reviewer, we added further details regarding mortality into our Material and Methods, Results and Discussion sections. We address the reviewer’s concerns in detail as the following.
Major comment #1: Referring to “The overall mortality as well as the 6-hour and 24-hour mortality were increased in group 2 patients with combined trauma.” The overall mortality should be described in more detail.
We thank the reviewer for raising this aspect. To state the overall mortality more precisely, we added further explanation to our Material and Methods and Results sections (see below). Here, we now explain that the overall mortality includes all patients deceasing during the primary hospital stay, i.e., during the emergency room phase, the primary operation phase, during their stay on ICU or on normal ward. The overall mortality does not include patients that passed away after discharge from hospital because the data basis of the TraumaRegister DGU® ends at this time point. We agree with the reviewer that our explanation regarding this variable was imprecise and thus added the missing details to the revised version of our manuscript. In detail, we now state the following:
“The overall mortality describes all patients that deceased during the primary hospital phase. Patients that passed away after discharge from hospital are not included, as the data basis from the TR-DGU ends at this time-point.” (ll. 101-103).
“The overall mortality (i.e., comprising all patients that deceased during the hospital phase) as well as the 6-hour and 24-hour mortality were increased in group 2 patients with combined trauma (Table 2).” (ll. 153-155).
Major comment #2: Referring to “The overall mortality as well as the 6-hour and 24-hour mortality were increased in group 2 patients with combined trauma.” The mortality after 24-hour seems to be decreased in group 2 patients with combined trauma, which was opposite to the overall mortality, 6-hour and 24-hour mortality. Did long term correlation and short term correlation were opposite? This is important and interesting.
We thank the reviewer for this interesting point. As explained above, our analysis distinguished between the 6-hour mortality, the 24-hour mortality and the overall mortality, i.e., including also patients that passed away after 24 hours in the hospital and the time of discharge. However, we do not present a mortality group including only patients dying after 24 hours which might describe the long-term aspect mentioned by the reviewer. To address the aspect raised by the reviewer, we compared the proportional mortality increase in the three groups (6-hour, 24-hour, overall). Interestingly, mortality within 6 hours was increased by more than 100% in patients with combined trauma (from 1.5% to 3.2%), whereas it was only increased by ~50% within 24 hours (from 5.6% to 7.9%, see Table 2). This indicates a relative reduction in mortality during early time course. In addition, the overall mortality (including also the later decedents as explained above) was only increased by ~20% in patients with combined TBI and femoral fracture (from 13.4% to 16.1%, Table 2). From our perspective, this comparison between short-term and long-term mortality is not exactly the opposite but shows a reduction during time course after trauma which is interesting to mention. Thus, we included a new paragraph into our Results and Discussion sections addressing this aspect:
“Interestingly, whereas the 6-hour mortality was increased by more than 100% in patients with combined trauma, the 24-hour mortality was only increased by ~50% (Table 2).” (ll. 155-157).
“Interestingly, the 6-hour mortality was increased by more than 100% in patients with combined trauma (from 1.5% to 3.2%), whereas the 24-hour mortality was only increased by ~50% (from 5.6% to 7.9%, Table 2). This indicates a relative reduction in mortality during early time course after trauma.” (ll. 297-299).

Reviewer 2 Report
Good paper. A huge data set and group of patients that were analyzed.
Just a few things to alter.
1) Throughout the paper the word "Conservative" is used. A better term is "Nonoperative". Please change throughout the manuscript and for the Tables and Figures.
2) In the Tables, embolden the significant numbers. that way they stand out for the reader.
3) There was no mention of the entity of Fat Embolism. This certainly occurs in this group of patients. Where is it found? This reviewer would like it more easily seen in this study. This reviewer notes that a TBI is bad, you have found that a TBI and a femur fracture is worse but this reviewer has seen that a TBI and femur fracture with fat embolism is very bad! You must have seen some of these patients but they are no where to be found. There should be some mention of this entity in this paper in the Discussion and maybe a way that you looked for it in the Methods and then if you found it in the Results. This is a big point as Fat Embolism is a very nasty entity when it is loaded upon a TBI.
Author Response
Good paper. A huge data set and group of patients that were analyzed.
We thank the reviewer for the careful assessment of our manuscript and helpful comments to improve our work. Following the reviewer’s recommendations, we revised our manuscript changing some wording within the text, highlighting significant numbers within Table 1 and 2, and, furthermore, adding a new section into our Discussion including a new reference (Davis et al., 2020, see below) addressing the aspect of fat embolism in TBI patients. We respond to the reviewer’s concerns as explained in detail below.
Just a few things to alter.
1) Throughout the paper the word "Conservative" is used. A better term is "Nonoperative". Please change throughout the manuscript and for the Tables and Figures.
We thank the reviewer for raising this aspect. Following the reviewer’s concern, we changed the wording within our manuscript from “conservative” to “non-operative; abbreviated: non-op” (ll. 113, 162, 163, 188 and Figure 3).
2) In the Tables, embolden the significant numbers. that way they stand out for the reader.
We thank the reviewer for raising this aspect. According to this suggestion, we emboldened the significant numbers in Table 1 and 2 (ll. 198-242).
3) There was no mention of the entity of Fat Embolism. This certainly occurs in this group of patients. Where is it found? This reviewer would like it more easily seen in this study. This reviewer notes that a TBI is bad, you have found that a TBI and a femur fracture is worse but this reviewer has seen that a TBI and femur fracture with fat embolism is very bad! You must have seen some of these patients but they are nowhere to be found. There should be some mention of this entity in this paper in the Discussion and maybe a way that you looked for it in the Methods and then if you found it in the Results. This is a big point as Fat Embolism is a very nasty entity when it is loaded upon a TBI.
We thank the reviewer for alterting us to this aspect. We agree that fat embolism is an important entity in TBI patients and worth investigating in this context. Unfortunately, the data base used in this manuscript, the TraumaRegister DGU®, does not hold sufficient information about this type of embolism, thus, detailed analysis in our patient cohort was not possible. The register retrieves information about “clinically relevant thrombo-embolic events” and if any occurred, distinguishes between i) myocardial infarction, ii) pulmonary embolism, iii) deep venous thrombosis (DVT) of lower extremity, iv) apoplexy, stroke, and v) other thrombo-embolic events (see TraumaRegister DGU®, Standard sheet V2020, https://www.traumaregister-dgu.de/fileadmin/user_upload/TR-DGU_-_Standard_form_English.pdf). The last category might include fat embolism; however, this is not further distinguishable from our perspective. Thus, we summarized all type of thrombo-embolic events as outcome variable (Table 2), which might include fat embolism as well. Nevertheless, we absolutely agree that this is an important aspect and added a paragraph into our Discussion section addressing this point:
“In addition, we found no difference in the rate of thromboembolic events between both groups, although, we consider especially fat embolism an important entity in TBI patients [44]. As the TR-DGU does not retrieve detailed information about this type of thromboembolic complication, analysis was not applicable, even though relevant to investigate in future work.” (ll. 324-327).
Literature:
[44] Davis, T., et al., The intersection of cerebral fat embolism syndrome and traumatic brain injury: a literature review and case series, Rev. Brain Inj. 2020. 34(8): p. 1127-1134. doi: 10.1080/02699052.2020.1776898.

Round 2
Reviewer 1 Report
The Authors have revised adequately the manuscript which now addressed all the comments.
Author Response
We thank the reviewer again for the careful revision of our manuscript and have no further comments.
Reviewer 2 Report
Good revision - no deficiencies noted.
Author Response
We thank the reviewer for careful assessment of our work and have no further comments.